# VIDEO-CSR: COMPLEX VIDEO DIGEST CREATION FOR VISUAL-LANGUAGE MODELS

## ABSTRACT

We present a novel task and human annotated dataset for evaluating the ability for visual-language models to generate captions and summaries for real-world video clips, which we call **Video-CSR** (**C**aptioning, **S**ummarization and **R**etrieval). The dataset contains 4.8K YouTube video clips of 20-60 seconds in duration and covers a wide range of topics and interests. Each video clip corresponds to 5 independently annotated captions (1 sentence) and summaries (3-10 sentences). Given any video selected from the dataset and its corresponding ASR information, we evaluate visual-language models on either caption or summary generation that is grounded in both the visual and auditory content of the video. Additionally, models are also evaluated on caption- and summary-based retrieval tasks, where the summary-based retrieval task requires the identification of a target video given *excerpts* of a given summary. Given the novel nature of the paragraph-length video summarization task, we perform extensive comparative analyses of different existing evaluation metrics and their alignment with human preferences. Finally, we propose a foundation model with competitive generation and retrieval capabilities that serves as a baseline for the Video-CSR task. We aim for Video-CSR to serve as a useful evaluation set in the age of large language models and complex multi-modal tasks.

## 1 INTRODUCTION

With billions of active users on video content platforms such as YouTube and TikTok, there has been an unprecedented need for automated complex video understanding. Classically, video understanding has focused on captioning and/or retrieval tasks on short videos with brief sentence-long captions. The concise nature of both the videos selected and captions labeled has partly been the result of model limitations, where detailed and nuanced multi-sentence video descriptions have not been possible with lightweight text decoders. With the recent leaps in large language models (LLMs), however, vision-language models (VLMs) now have the opportunity to tap into the immense natural language capabilities of models such as LLaMA (Touvron et al., 2023a;b) and ChatGPT(Ouyang et al., 2022; OpenAI, 2023). With tens to hundreds of billions of parameters, these LLMs are able to write entire essays with details and poise that mimic human to an unprecedented extent. With video conversational models such as Video-LLaMA(Zhang et al., 2023), Video-ChatGPT(Maaz et al., 2023) and VideoChat (Li et al., 2023b) claiming to be able to generate detailed and fine-grained descriptions of video inputs, we believe the time is ripe for an evaluation benchmark that matches the capabilities of modern VLMs powered by LLMs.

In the current work, we focus on videos with multi-shot compositions containing diverse information streams such as dialogues, background music, and complex visual sequences. We developed **Video-CSR**, a novel task and dataset for long form **Video C**aptioning, **S**ummarization and **R**etrieval. This new, multi-modal dataset contains 4.8K video clips carefully selected from previously published YouTube-based video datasets (YouTube-8M (Abu-El-Haija et al., 2016) and YT-Temporal-1B (Zellers et al., 2022)) that integrates visual with auditory information. Over months, a team of 24 human annotators (college and graduate level students) created 5 short captions (1 sentence each) and 5 long summaries (3-10 sentences) for each video clip, resulting in a rich and comprehensive human-annotated dataset that serves as a robust ground truth for subsequent model training and evaluation (See Figure 1 for example).

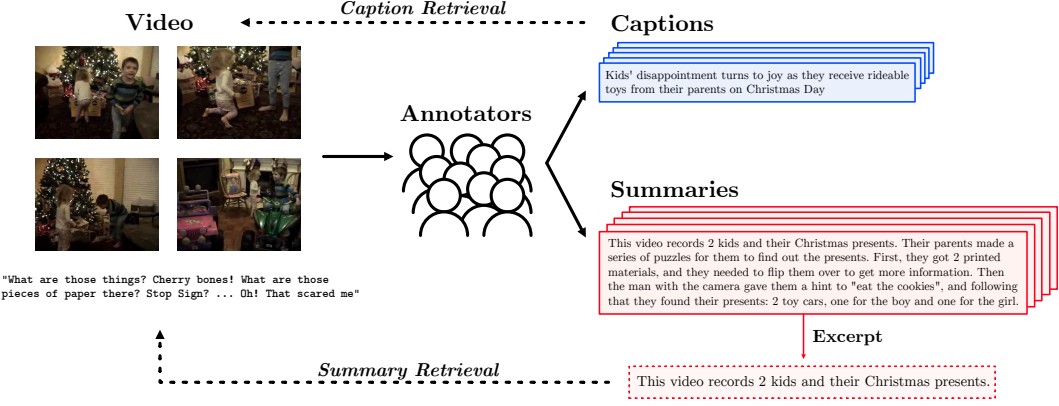

Figure 1: **Example of Video-CSR dataset.** For each video, 5 captions and 5 summaries are independently annotated based on both visual and auditory information of the selected videos. Text-to-video retrievals from video summaries are performed by randomly sampling single-sentence excerpts.

As opposed to short video captions where N-gram based metrics such as CIDEr (Vedantam et al., 2015) offer good alignment with human preferences, it is not immediately apparent as how to evaluate long form video summarization and retrieval. By comparing N-gram-based metrics and model-based metrics (e.g. BLEURT Sellam et al. (2020), BERTScore Zhang et al. (2020)) to human preferences, we found that model-based metrics are better suited for long form summarization tasks.

Finally, we evaluate different types of VLM architectures on our test set, aiming to provide a comprehensive landscape of what is currently feasible and effective on our task. Specifically, we compare Video-LLaMA, a video-language model with frozen LLMs, to a novel resource efficient end-to-end foundation model developed based on VideoCoCa (Yan et al., 2023). Termed SimCSR, our model aims to provide a **Sim**ple yet effectively end-to-end model that covers all three tasks at hand. We find that while Video-LLaMA performs better on generation tasks, it suffers from hallucination and is unable to be applied for video retrieval. In contrast, SimCSR offers comparable generation performance and excels at video retrieval based on both video captions and summaries.

Our contributions are summarized as follows:

- We introduce a new dataset of human annotated video caption (1 sentence) and summaries (3-10 sentences) to gauge the ability of VLMs to perform long form summary of video content. To the best of our knowledge, Video-CSR is the first comprehensive human-annotated evaluation dataset for long form video summaries.

- We compare different evaluation metrics for long form video summarization task and find that model-based metrics offer better alignment to human preference.

- We develop a baseline foundation model (SimCSR) that integrates visual, auditory and textual modalities for both generation and retrieval tasks.

## 2 RELATED WORK

The endeavor to understand and provide textual descriptions of video content has been the subject of numerous research initiatives. Here we briefly review recent models and datasets relevant to Video-CSR.

**Video-Language Models**  In the video-language model landscape, two model architectures are prevalent. The first category encompasses the *end-to-end trainable models*, under which our Sim-CSR model belongs. End-to-end models such as BLIP (Li et al., 2022) and VideoCoCa (Yan et al., 2023) are designed to learn representations from both videos and text simultaneously, without any frozen modules. These models are often more parameter-efficient than models equipped with frozen LLMs, which we describe below.

Pioneered by models including BLIP-2 (Li et al., 2023a), *models with frozen modules* have dominated the video-language scene since the introduction of powerful LLMs like ChatGPT (Ouyang et al., 2022). Models like BLIP-2 (Li et al., 2022), Video-LLaMA(Zhang et al., 2023), Video-ChatGPT(Maaz et al., 2023), and VideoChat (Li et al., 2023b) integrate pre-trained frozen linguistic components complemented by additional trainable components, often a lightweight visual backbone. Effectively, such models take advantage of the natural language abilities of LLMs by providing soft prompts encoded by a lightweight trainable mutlimodal adaptor. As LLMs are capable of both consuming and generating texts with hundreds if not thousands of words, models in this category are often capable to generate video summaries. We hope that our Video-CSR benchmark will contribute the continued advancements in these powerful video-language models.

**Video-Language Datasets**   Datasets in this domain can be broadly categorized based on their domain specificity and downstream tasks. Under *domain specificity*, datasets like MSVD (Chen & Dolan), MSR-VTT (Xu et al., 2016), YouTube-8M (Abu-El-Haija et al., 2016), YT-Temporal-1B (Zellers et al., 2022), HD-Vila-100M (Xue et al., 2022) provide a panoramic view of diverse video content, fostering a comprehensive model understanding. In contrast, datasets such as How2 (Sanabria et al., 2018) YouCook2 (Zhou et al., 2017) and HowTo100M (Miech et al., 2019) predominantly focus on instructional content.

In terms of *task orientation*, open domain datasets mentioned above are often focused on video-to-text generation and retrieval tasks. In contrast, datasets such as Kinetics-700 (Carreira et al., 2022) and ActivityNet (Krishna et al., 2017) focus on more specialized downstream applications such as activity detection.

As our dataset is designed to primarily gauge the ability for models to accurately capture a balanced understanding of overall content and details in a given video, we focused on open domain videos and generation/retrieval tasks.

## 3   VIDEO-CSR DATASET

In this section, we describe the procedure which with Video-CSR was constructed and how generation and retrieval task performances are evaluated on Video-CSR.

### 3.1   EVALUATION DATASET

The dataset utilized in this study is an amalgamation of YouTube videos, which were source from two previously available large-scale video datasets: YouTube-8M and YT-Temporal-1B.

The selection of videos for human annotation was focused on English videos with high quality and diversity, and saw one significant evolution during the course of the annotation process. Refer to Figure 5 in Appendix A for examples of relevant metadata information used during the video selection process.

**First Phase: 2.3K Videos**   In the **first phase** of the data curation process, videos are selected from YouTube-8M and YT-Temporal-1B datasets based solely video metadata with the following criteria:

- Video title, description and subtitles (if applicable) must be primarily in English;

- Video must contain Chapter information, which is video keyframe information provided by video uploaders;

- Video clips, when segmented based on chapter information, should be between 20 to 60 seconds.

We find that of all videos in the YouTube-8M and YT-Temporal-1B datasets, roughly 1% satisfied our constraint. All accounted for, around 2.3K video segments were curated following this procedure, which form the **first phase** of our data annotation process. Note that the 100K training dataset mentioned later in this paper was curated in tandem with the **first phase** evaluation dataset, as such the distribution of our training dataset aligns best with this portion of the test set (see Section 3.4 for more information).

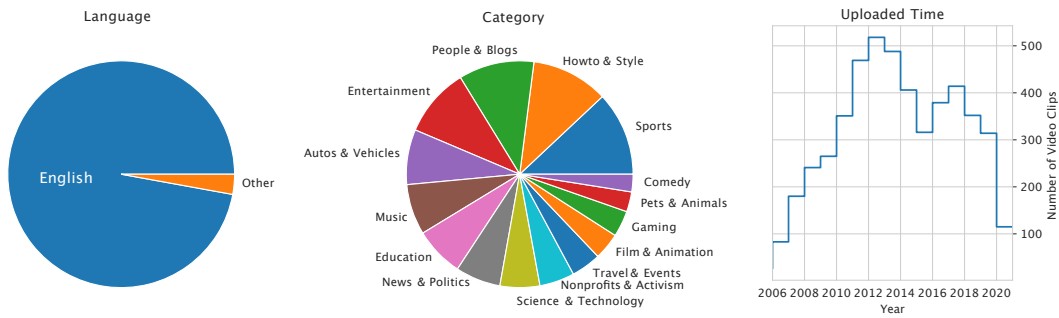

Figure 2: **Diversity of Video-CSR dataset.** Our dataset contains mainly English videos covering a diverse range of topics uploaded across the past 17 years.

**Second Phase: 2.5K Videos**   In the **second phase** of the data annotation process, we adjust the criteria to favor videos for which visual-grounding is *necessary* for accurate annotation. In particular, previous selection criteria (most significantly, the requirement for "Chapter" information) led to a bias towards News and Instruction type videos, for which speech information contents were dominant.

As a result, human annotators often heavily favored, for example, the *content* of the News articles being broadcasted over the actual *visual setting* of the broadcasting room. This has the undesired consequence that visual-language models with strong language capabilities but weaker visual grounding can potentially have better performance than the more visually-grounded counterparts. Additionally, to avoid videos that are montages of static images, we apply an additional frame-embedding based filter to select videos with high inter-frame variability. The video selection process is as follows.

1. The audio content of each video is first processed by `Whisper-Base` (Radford et al., 2022) to generate automatic speech recognition (ASR) content, followed by an entropy computation, where only videos with entropy lower than 4.2 are kept.

2. The visual content of each video is evaluated by uniformly sampling 8 frames and computing embedding of each frame using CLIP (Radford et al., 2021); $L_2$ distances between embeddings of neighboring frames are computed and averaged, where only videos with average inter-frame $L_2$ distance above 5.5 are kept.

3. Video title, description and subtitles (if applicable) must be primarily in English.

4. Instead of "Chapter" information, videos are segmented using keyframes detected via TransNet (Souček et al., 2019). Only segments that satisfy the 20-60 second duration requirement are kept.

Selected video clips are filtered manually during the annotation process where annotators were provided the option to discard a given video if it is deemed of poor quality: non-English or does not contain sufficient information content for summarization in 3-10 sentences. We find that roughly *20%* of automatically selected videos were filtered by human annotators.

The months-long annotation process is divided into multiple rounds, with each round covering 500-1500 videos. After each round of annotation, 20% of videos are randomly selected for quality control independent of the original annotators. If systematic problems are detected in a batch of annotations, the entire batch is returned to annotators for revision before going through another round of quality control. In later rounds, as quality of annotation stabilized, the percentage of videos selected for independent quality control is adjusted downwards to a minimum of 7.5%. This process is repeated until the batch at question is deemed of satisfactory quality, and every batch went through at least one round of revision.

As shown in Figure 2, the final 4.8K evaluation dataset contains videos primarily in English, and covers a wide range of topics and interests. The statistics of the videos and annotations are shown in Table 1 and Figure 6 in Appendix B.

Table 1: **Average values of data statistics in Video-CSR evaluation set.** Refer to Figure 6 in Appendix B for detailed distribution.

| Duration [Seconds] | Words in Caption | Words in Summary | Words in ASR / Second |
|---|---|---|---|
| 22.3 | 12.71 | 62.93 | 1.66 |

## 3.2 TASKS AND EVALUATION METRICS

**Video Captioning Task**    For the short video captioning task, our evaluation methodology adheres to classic approaches adopted in prior works such as MSR-VTT (Xu et al., 2016). We report commonly used N-gram based evaluation metrics including BLEU (Papineni et al., 2001), ROUGE-L (Lin, 2004), METEOR (Banerjee & Lavie, 2005) and CIDEr (Vedantam et al., 2015) for all our results to gauge the quality of the model-generated captions. These metrics have proven effective for evaluating the lexical overlap and syntactic structure in brief captions, which are relatively straightforward and independent of language models.

**Video Caption Retrieval Task**    For evaluating the efficacy of models in the short video retrieval task, we follow the classic retrieval accuracy metrics at different levels of granularity: Recall @1, @5, and @10.

**Video Summarization Task**    In contrast to video captioning, the task of evaluating long video summaries presents distinct challenges. We found that ranking of annotations via N-gram based and model-based (BLEURT (Sellam et al., 2020)) metrics differ more significantly for video summaries as opposed to video captions. By comparing to human preferences, we found that model-based metrics such as BLEURT have a better agreement to human preferences (see Section 3.3 for more details) and therefore serve as our primary method of evaluation for video summarization.

**Video Summary Retrieval Task**    In addition to the conventional retrieval tasks, we introduce a new evaluation scheme for long caption summaries. Specifically, the text-to-video retrieval is performed by sampling multiple sentences from the given video summary, the retrieval performance of all sampled sentences were averaged to give the final accuracy. Note that only sentences with more than 5 words were used to ensure that excerpts of video summaries contain sufficient information for retrieval. This task mimics the common scenario where viewers may desire to search for videos based on memories of partial information. For this task, we also report Recall @1, @5, and @10.

## 3.3 ALIGNMENT OF EVALUATION METRICS TO HUMAN PREFERENCE

Given the novel nature of long form video summarization task, we sought to compare the alignment of different evaluation metrics to human preferences.

To start, we computed the correlation between different evaluation metrics on both captioning and summarization tasks. Using one annotated response as prediction and all other responses as ground truths, we computed N-gram based (CIDEr) and model-based (BLEURT) metrics for both captioning and summarization tasks. We observe that across all annotators and all videos, the correlation between CIDEr and BLEURT metrics is 0.58 for captioning task and 0.43 for summarization task. Importantly, for captioning, all N-gram metrics have lower than 0.62 correlation to BLEURT while the minimum correlation between N-gram metrics is 0.75. This result suggests that while N-gram based metrics may offer consistent metric for captioning task, they differ significantly from model-based metrics in both short and long form video summarization tasks.

To determine the most suitable metric for evaluating long form summarization task, we compared ranking by metrics to that of human preference. We selected 20 videos from annotated responses 1 and 5 as predictions, and computed the corresponding captioning and summarization metrics against annotations 2,3,4. The videos were randomly selected from a subset of videos where ranking produced by CIDEr and BLEURT metrics differ. Note that only annotations 2,3,4 are considered as ground truths to ensure that the same set of ground-truth annotations are used for a fair comparison between responses 1 and 5. We manually ranked the responses by perceived quality and relatedness

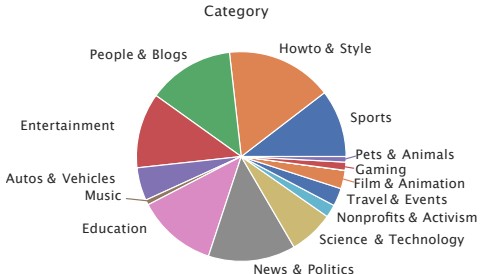 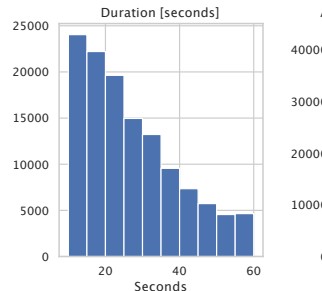 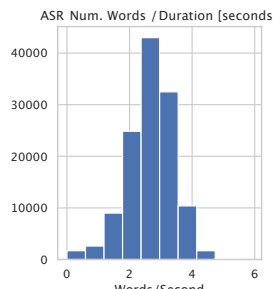

Figure 3: **Diversity of Training dataset.** Our training dataset contains captions and summaries for 100K ASR-rich video segments. Note that as opposed to test dataset in Figure 2, the ASR Number of Words normalized by video duration does not have a significant concentration around 0, indicating that all videos in the training dataset contains a significant amount of ASR information.

to the video content, and compared human ranking to ranking based on metrics. We observe a 56% alignment of human ranking to ranking by CIDEr score and a 67% alignment to that by BLEURT.

It is worth noting that while the difference in human alignment between BLEURT and CIDEr appears significant, it is biased by preferences by human labelers and limited by the number of videos sampled during comparison. Additionally, averaged across many videos, all metrics align mostly with human preferences. Nevertheless, the difference between CIDEr and BLEURT for video summaries point to the differences between semantic and lexical similarities when evaluating paragraph-length textual descriptions. We chose BLEURT as our metric for video summarization task but it remains unclear how better alignment to human preferences should be evaluated and implemented for complex multi-modal tasks with long form text generation.

### 3.4 TRAINING DATASET

In addition to the 4.8K evaluation dataset, we also prepared a training dataset with 100K video clips whose captions and summaries were generated using metadata information (e.g., title, description, category, ASR, etc.). The training dataset is comprised of 100,000 videos segments, selected via the same procedure as in the **first phase** of evaluation dataset curation (see Section 3.1). Note while due to time constraint, we were not able to collect another training set based on the selection criteria in the **second phase** of evaluation dataset development.

Similar to videos in the evaluation dataset, videos in the training dataset were segmented based on the video chapter information. A prompt template was then used to create queries for an LLM to generate 5 captions and summaries for each video (see Appendix D). To exercise control over the text generation process and maintain consistency, each prompt was prefixed with "This video". At the time of dataset creation, the `gpt-3.5-turbo` (Ouyang et al., 2022) model was observed to have significant more issues related to hallucinations as compared to `text-davinci-003` (Brown et al., 2020), thereby motivating the choice of the seemingly less powerful but more "reliable" `text-davinci-003` for our specific requirements.

It's worth noting that, as mentioned in the description of the **second phase** of evaluation dataset development in Section 3.1, training dataset selected following criteria in the **first phase** led to a skewed distribution towards ASR-rich videos (see Figure 3). Due to time constraints, we could not diversify this training dataset further and acknowledge this as a limitation. Future work will aim to rectify this skewness by creating a more balanced training dataset.

### 4 EXPERIMENTS

In this section, we describe both human performance and model performance on Video-CSR evaluation set. To comprehensively evaluate the capabilities and limitations of various architectures for video captioning and retrieval, we consider two distinct types of models, each with its own set of advantages and drawbacks.

Table 2: **Human Performance for Different Tasks.** Annotation from each annotator is used as *prediction* and computed against *ground truth* results from all other 4 annotators. The overall metrics are then aggregated via Average and Minimum.

| | Video Caption | | | Video Summary | | | |
|---|---|---|---|---|---|---|---|
| | CIDEr | ROUGE-L | METEOR | CIDEr | ROUGE-L | METEOR | BLEURT |
| Average | 0.3836 | 0.3747 | 0.273 | 0.355 | 0.338 | 0.257 | 0.6234 |
| Minimum | 0.3319 | 0.3319 | 0.240 | 0.286 | 0.230 | 0.240 | 0.5860 |

Table 3: **Same annotator re-labeling summaries of the same video twice is equivalent to the same video summary labeled by different annotators.** 150 randomly selected captions and summaries were relabeled by the same annotator at least 1 month apart from the original annotation. We observe that annotations by the same annotator 1 month apart have similar level of discrepancy as compared to annotations by different annotators.

| Task | Ground Truth | Metrics | |
|---|---|---|---|
| | | CIDEr | BLEURT |
| Caption | Other Annotators | 0.3111 | - |
| | Same Annotator 1 Month Apart | 0.4128 | - |
| Summary | Other Annotators | 0.5034 | 0.5960 |
| | Same Annotator 1 Month Apart | 0.3843 | 0.5630 |

## 4.1 HUMAN PERFORMANCE

For a comprehensive understanding of model capabilities, it is imperative to establish a benchmark for human performance on the tasks at hand. To this end, we employ a strategy wherein one human annotation is used as the 'prediction', while the remaining four annotations serve as the 'reference' set. The aggregated results across annotators for both captioning and summarization tasks are shown in Table 2. For detailed metrics of each annotator, see Table 7 and Table 8 in Appendix C.

Furthermore, we conducted an additional layer of analysis by revisiting the stability of human annotations over time. Specifically, for the same video annotated by the same annotator more than one month apart, we compared the similarity between these two annotations. These similarity metrics were then contrasted with the similarity of the initial annotation when compared to those produced by the other human annotators. A shown in Table 3, compared to other annotators, same annotator is able to achieve a higher CIDEr score for captioning task but a lower CIDEr and BLEURT score for summarization task. This indicates that annotators are able to maintain a much higher consistency for captioning task over summarization task. In fact, comparing BLEURT score for summary results in Table 3 indicates that annotation by the same annotator 1 month apart is equivalent to annotation by different annotators. This exercise provides not only an internal measure of human consistency but also establishes the subjective difficulty of the long form video summarization task. It is therefore crucial to have multiple (at least 5 in our case) summary annotations for each video to ensure the diversity of ground truths.

## 4.2 TRAINABLE VISUAL ENCODER WITH ADAPTOR ON FROZEN LLMS

Recently, many methods combining visual encoders with frozen LLMs have emerged. Pioneered by models including BLIP-2, recent additions to this category of models include Video-LLaMA, which utilizes both visual and auditory information from videos and proposes separate branches for each modality. As Video-LLaMA integrates both visual and auditory modalities, it serves as a good reference model for Video-CSR.

Piggybacking off of the natural language power of LLMs, models like Video-LLaMA offer seemingly impressive multi-modal captioning and summarization performances. Indeed, as shown in Table 5 that will be presented later, Video-LLaMA is able to achieve a BLEURT score of 39.3, which is higher than our end-to-end model (BLEURT 31.4).

Table 4: **Degree of hallucination of model with LLM compared with SimCSR model.** Note that the 100K Video-CSR training dataset is in fact out-of-distribution with the evaluation dataset (see Section 3.4). As such evaluation of both models on Video-CSR evaluation dataset may be considered as zero-shot.

| Model | Degree of Hallucination | | | Parameters | Finetuning Dataset |
|---|---|---|---|---|---|
| | No | Moderate | Severe | Trainable/Total | |
| Video-LLaMA | 26 % | 40 % | 34 % | 50M / 7B | VideoChat (11K) |
| SimCSR | 60 % | 26 % | 14 % | 750M / 750M | Video-CSR Train Set (100K) |

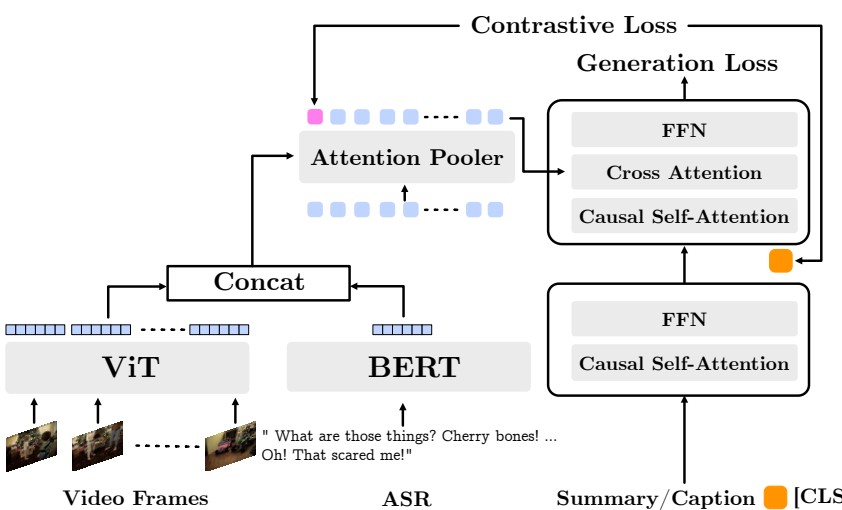

Figure 4: **SimCSR Model Architecture.** Our End-to-End model combines VideoCoCa model architecture with an additional ASR encoder. Frame-level embeddings of VideoCoCa and ASR embeddings are concatenated before passing through the Attention Pooler. Note that the contrastive loss is computed using the first output embedding of the Attention Pooler.

However, we noted that video summaries generated by Video-LLaMA often suffers from the problem of hallucination. To quantify such problem, we randomly selected 20 summaries generated by Video-LLaMA and our end-to-end model SimCSR (see next section), and evaluated the degree of hallucination in their responses. We employ a random sampling technique to select 25 sentences from the generated outputs, subsequently gauge their perceptual illusion intensities through different person individually. The criteria employed in our assessment are delineated in Table 9 of Appendix F. As shown in Table 4, Video-LLaMA suffers from significantly higher degree of hallucination, which may be ascribed by rich prior knowledge of the large model.

### 4.3 SIMCSR: END-TO-END FOUNDATION MODEL

Built upon the architecture of VideoCoCa (Yan et al., 2023), we developed an end-to-end trainable foundation model that includes both a visual encoder and ASR encoder, termed SimCSR (see Figure 4 for model architecture). As no open source VideoCoCa implementation was available, we implemented SimCSR from scratch following the **Attention Pooler** type model described in the VideoCoca manuscript.

Following CoCa (Yu et al., 2022) and VideoCoCa, the text encoder takes in Caption or Summary as input where a special [CLS] token is suffixed to all input sequences. The text encoder is evenly divided into unimodal (bottom) and multi-modal parts, where the output encoding of the [CLS] token by the unimodal encoder is used to compute contrastive training objective against other modalities.

On the visual encoder side, 8 frames are uniformly sampled from each video and encoded independently by ViT. The output visual encoding of each frame is concatenated to form the overall representation of the visual information in an input video.

Table 5: **Results for video to text generation of SimCSR on Video-CSR evaluation dataset.** RG-L refers to ROUGE-L.

| Model | Video Summary | | | Video Caption | | | |
|---|---|---|---|---|---|---|---|
| | BLEURT | RG-L | CIDEr | BLEU-4 | METEOR | RG-L | CIDEr |
| Video-LLaMA | **39.3** | 19.2 | 2.1 | - | - | - | - |
| SimCSR w/o ASR | 29.8 | 21.0 | 5.3 | 2.1 | 10.2 | 18.1 | 6.6 |
| SimCSR | 31.4 | **22.7** | **7.1** | **2.8** | **12.0** | **19.9** | **8.1** |

Table 6: **Results for text to video retrieval of SimCSR on Video-CSR evaluation dataset. R** refers to Recall. Note that Video-LLaMA is omitted from this comparison since retrieval is not supported.

| Model | Caption Retrieval | | | Summary Retrieval | | |
|---|---|---|---|---|---|---|
| | R@1 | R@5 | R@10 | R@1 | R@5 | R@10 |
| SimCSR w/o ASR | 46.2 | 66.5 | 73.3 | 37.7 | 56.2 | 63.4 |
| SimCSR | **56.8** | **74.3** | **79.8** | **40.8** | **58.3** | **64.8** |

The visual encoder output is further concatenated with BERT encoding of the ASR information. The visual and ASR encodings are then integrated and compressed by the "Attention Pooler" module with 256 query tokens. The first output token of Attention Pooler is used to compute contrastive loss against encoding of `[CLS]` token mentioned above.

Following CoCa, the training objective is a combination of generation loss and contrastive loss. All parameters are initialized from the OpenCLIP (Ilharco et al., 2021) implementation of CoCa, except for the ASR encoder which was initialized from `BERT-base` (Devlin et al., 2019). Refer to Appendix E for training details. Results of the video-to-text generation and text-to-video retrieval on Video-CSR can be found in Table 5 and Table 6, respectively. We performed an ablation study on the ASR encoder, and found that the integration of ASR information significantly improves both retrieval and generation performances of our model. This is consistent with the observation that roughly 50% of the evaluation dataset contains videos rich in speech content.

## 5 CONCLUSION

In this paper, we introduce a new multi-modal dataset curated from a diverse range of YouTube videos, designed to advance the state of the art in long-form video summarization tasks. Through a carefully orchestrated annotation process involving multiple human annotators, multiple rounds of video selection and quality control, we ensure that the dataset is comprised of high-quality and diverse captions and summaries.

We show that while qualities of one-sentence captions can be accurately evaluated by N-gram, multi-sentence summaries require a more semantically aligned metric such as BLEURT. We note that model-based metrics can be, and often are, finetuned based on specific downstream tasks where they are applied, and we intend to keep refining our evaluation metric to better alignt with human preferences.

From the modeling standpoint, we compare the performance between visual-language models with frozen LLMs to smaller end-to-end models. We find that, despite their impressive language capabilities, models with frozen LLMs come with significant computational costs and can sometimes generate hallucinatory content. In contrast, our proposed model serves as a resource-efficient baseline capable of both generation and retrieval tasks, albeit with a limited ability at generating extremely long summaries. It remains unclear how generation, retrieval can be best balanced while maintainig a low level of hallucination.

As the field of video summarization and captioning continues to evolve, it is imperative that datasets and evaluation metrics keep pace. Our work aims to serve as a stepping stone in this direction, providing a balanced approach to video understanding.

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

## A  YouTube Metadata information

YouTube metadata information refers to all information related to an uploaded video except for the video and audio content. This includes information such as title, description, category, tags, asr, chapter, playlist, subtitles, etc. In figure 5, we highlight a few important metadata information that is relevant for video selection in our work.

We highlight "Chapter" information, which corresponds to keyframe information with human annotated segment subtitles. These chapter information naturally divide video into semantically meaningful chunks that we used to split longer videos into 20-60 second clips.

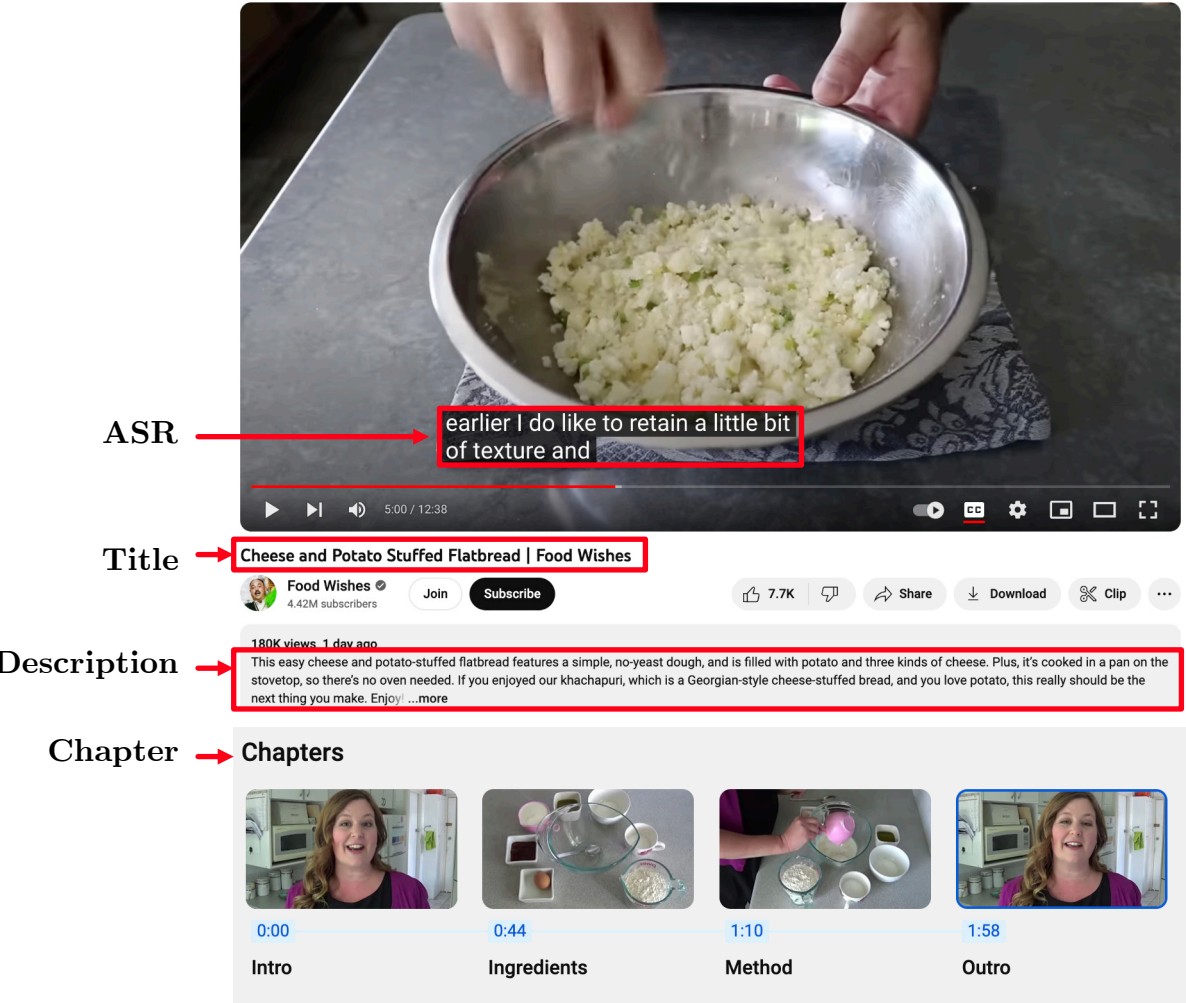

Figure 5: **Example of YouTube metadata information.** Note that only metadata that are relevant for video selection is shown. "Chapter" information contains keyframe information with headers provided by the video uploader.

## B  Detailed Statistics of Video-CSR Evaluation Dataset

Here we provide detailed statistics on the evaluation dataset. In particular, in second column from left, we show that the ASR content, when normalized by duration of video, demonstrates a clear bimodal distribution, corresponding to videos with high and low ASR content selected during the two phases of annotation process. Additionally, we observe that a significant portion of video sum-

maries are longer than 100 words, which is considerably longer than video annotations in previously available datasets.

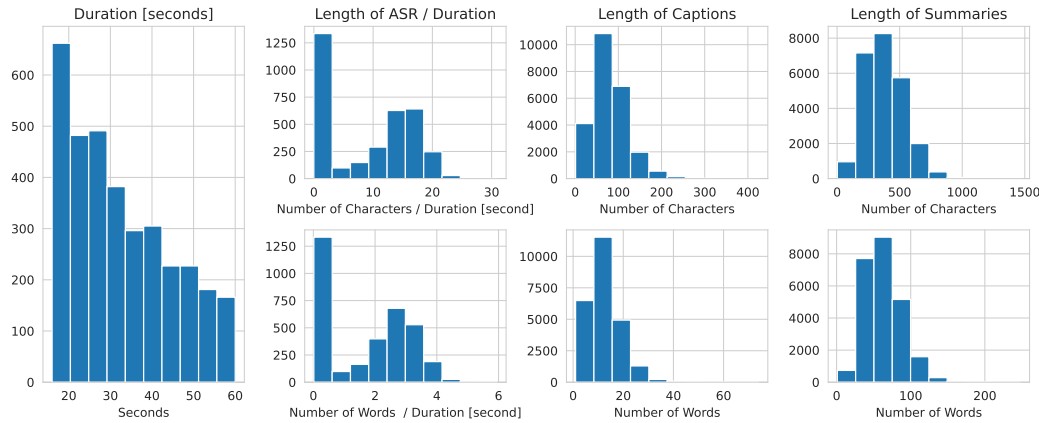

Figure 6: **Distribution of video duration, ASR content, length of captions and length of summaries of Video-CSR dataset.** The *Length of ASR* normalized by duration of video clip indicates that our dataset covers videos ranging from no speech content to high speech content. Note that number of words are calculated by counting the number of white-space-seperated character groups, and may contain punctuation.

## C   DETAILED GENERATION METRICS FOR EACH ANNOTATOR

Here we provide metrics comparing each annotator's caption & summaries to all other annotators. Each row in Table 7 and Table 8 correspond to metric evaluated using the given annotator's caption/summary as prediction and all others' as ground truth. We computed both Average and Minimum values for each metric for reference, where the minimum value indicates the lower-bound of human performance.

Table 7: **Human Performance of Caption Task.** Annotation from each annotator is used as *Prediction* and computed against Ground Truth results from all other 4 annotators. The overall metrics are then aggregated via Average and Minimum.

| Annotator | CIDEr | ROUGE-L | METEOR |
| --- | --- | --- | --- |
| 1 | 0.3769 | 0.2750 | 0.276 |
| 2 | 0.3849 | 0.3794 | 0.279 |
| 3 | 0.4219 | 0.3940 | 0.284 |
| 4 | 0.4025 | 0.3860 | 0.284 |
| 5 | 0.3319 | 0.3319 | 0.240 |
| Average | 0.3836 | 0.3747 | 0.273 |
| Minimum | 0.3319 | 0.3319 | 0.240 |

Table 8: **Human Performance of Summarization Task.** Annotation from each annotator is used as *Prediction* and computed against Ground Truth results from all other 4 annotators. The overall metrics are then aggregated via Average and Minimum.

| Annotator | CIDEr | ROUGE-L | METEOR | BLEURT |
|---|---|---|---|---|
| 1 | 0.375 | 0.344 | 0.258 | 0.6296 |
| 2 | 0.337 | 0.341 | 0.263 | 0.6305 |
| 3 | 0.416 | 0.354 | 0.267 | 0.6383 |
| 4 | 0.363 | 0.342 | 0.264 | 0.6327 |
| 5 | 0.286 | 0.307 | 0.230 | 0.5860 |
| Average | 0.355 | 0.338 | 0.257 | 0.6234 |
| Minimum | 0.286 | 0.230 | 0.240 | 0.5860 |

## D   PROMPT OF TRAINING DATA GENERATION

The following prompt template is used for generating video summaries in training set. Here `cc` refers to the video subtitles.

```
Please write a summary in 3 to 10 sentences that
accurately summarizes the video's content and captures
its essence based on the following information.

Title: {{ title }}|{{ chapter_title }}

Category: {{ category }}

Description: {{ description }}

Closed Captions: {{ cc }}

SUMMARY: This video
```

The following prompt template is used for generation video captions in training set. Here `cc` refers to the video subtitles.

```
Please write a 1-sentence caption that
accurately summarizes the video's content and captures
its essence based on the following information.

Title: {{ title }}|{{ chapter_title }}

Category: {{ category }}

Description: {{ description }}

Closed Captions: {{ cc }}

ONE-SENTENCE CAPTION: This video
```

## E   SIMCSR TRAINING DETAILS

SimCSR models are trained using 64 V100-32G GPUs with global batchsize of 256 (3 per card). Training 100K dataset for 5 epochs consumes roughly 5 hours.

Learning rate of SimCSR follows Linear Warmup with Cosine decay pattern, with warmup LR of 1e-7 (500 warmup steps), peak LR of 1e-5 and a minimum LR of 1e-6.

Both SimCSR and SimCSR w/o ASR models are trained with 8 uniformly sampled input video frames for 5 epochs.

# F  HALLUCINATION

To evaluate the degree of hallucination in model generation video caption and summaries, we first differentiate varying degrees of hallucination in generated video captions/summaries as shown in Table 9.

Table 9: Classification of degree of hallucination.

| Degree of Hallucination | Description |
|---|---|
| No | The generated description aligns with the video's content, albeit with potential omissions, ensuring the absence of any fabrication or inclusion of non-existent information within the video. |
| Moderate | The generated description generally conforms to the content of the video, but there are some fabricated information. |
| Severe | The generated description differs significantly from what is depicted in the video, and contains a substantial amount of content that is LLM's own imagination. |

We then randomly selected videos and compared model generated responses to ground truth annotations. Examples of moderate and severe hallucinations are shown below in Table 10.

Table 10: Examples of hallucinations.

| Degree of Hallucination | Ground Truth | Model Response |
|---|---|---|
| Moderate | in this video shows zimba serval, a 17 years old male serval cat. it is often found in a relaxed state amidst the lush green ferns. he shows no hesitation in approaching for delicious meat treats. | in the video, a zebra is seen relaxing in a fern garden. the speaker mentions that the zebra is being fed meat treats by someone. |
| Severe | the video features a man reading a book. the text is written in a foreign language, and the man is unable to read it. the video ends with a close-up of the man's face as he reads the text. | in the video clip, nicholas hoult is revealed as the star of an upcoming movie. the film is said to have had a significant impact on the creative vision of a renowned author, suggesting that it played a crucial role in shaping their imagination. |

