# OpenReview forum: "Video-CSR: Complex Video Digest Creation for Visual-Language Models"
_ICLR.cc/2024/Conference — ICLR 2024 Conference Withdrawn Submission_

### Official Review · Reviewer_Bzf5 · 2023-10-29

**Soundness:** 2 fair
**Presentation:** 1 poor
**Contribution:** 2 fair
**Rating:** 3
**Confidence:** 2

**Summary:**

This paper introduces a new dataset for evaluating video captioning, summarization and retrieval.


About the proposed dataset Video-CSR:
It contains 4.8K YouTube video clips of 20-60 seconds duration with 5 independently annotated captions (1 sentence) and summaries (3-10 sentences) per video.
The videos cover diverse topics and have both visual and speech content.


The findings based on evaluation results of Video-CSR:
They compare different evaluation metrics for the long-form summarization task and find model-based metrics like BLEURT have better correlation to human rankings than n-gram metrics.
Two model architectures are evaluated: Video-LLaMA which uses a frozen large language model, and an end-to-end "SimCSR" model they propose. Video-LLaMA has good text generation but suffers from hallucination, while SimCSR balances generation and retrieval without hallucination.

**Strengths:**

The paper do brings some insights on how to construct a carefully annotated dataset for multiple multi-modality tasks.

**Weaknesses:**

The major issue of this paper is the dataset and evaluation size.
With 4.8K test videos, metrics can have high variance. Larger scale evaluation would provide more confidence.
Also mentioned in the paper, some evaluation is based on only tens of samples:

Sec 3.3 … we compared ranking by metrics to that of human preference. We selected 20 videos from annotated responses…

Sec 4.2 … we randomly selected 20 summaries generated by Video-LLaMA and our end-to-end model SimCSR (see next section), and evaluated the degree of hallucination in their responses…


The motivation behind the joint evaluation of captioning and summarization is unclear. The authors evaluate both capabilities on the same relatively small dataset of 4.8k videos. However, captioning and summarization could be evaluated separately on larger datasets specifically tailored for each task. For example, captioning could be evaluated on existing datasets like MSR-VTT and summarization on long-form video datasets (e.g. synopsis). The rationale for creating a smaller joint evaluation benchmark rather than leveraging larger separate datasets for each task is not sufficiently justified. The authors could expand on their motivations for joint evaluation rather than independent testing on larger captioning and summarization datasets. As it stands, the decision to combine the tasks may limit the scale and focus of the evaluation.

**Questions:**

Please elaborate on my concerns as listed in Weaknesses:
- Evaluation set size
- Motivation of joint evaluation

---

> ### Author Response · Authors · 2023-11-22
>
> We thank the reviewer for the comment. We plan on extensively revising our manuscript with a larger evaluation dataset (10K) and more extensive benchmarking and will be withdrawing our submission as a result.
> However, we'd like to address a few points raised by the reviewer for clarification.
> 1. Extensiveness of the dataset: our evaluation dataset (4.8K videos with 5 human-annotated captions and summaries per video) is similar to or larger in size than similar datasets in the literature (e.g. MSVD 2000 video segments, MSRVTT 3K video clips) albeit with significantly longer video durations (upwards of 60 seconds as opposed to ~10 seconds in MSRVTT).
> 2. Limited human evaluation for model hallucination: we acknowledge that 20 randomly selected samples is limited in scale and are working on increasing the scale of this evaluation.
> 3. Joint Evaluation: the reviewer seems to suggest that evaluating model's ability to generate both short and long textual descriptions on a common set of videos limit the scale of the evaluation dataset. We'd like to clarify that the fundamental limitation of the scale of the benchmark is due to the fundamental challenge of human annotation. Additionally, we argue that providing a benchmark that supports both evaluation of short and long video descriptions does not limit the application of the video-CSR dataset to only either one of the two applications.

---

### Official Review · Reviewer_2rQ2 · 2023-10-31

**Soundness:** 3 good
**Presentation:** 2 fair
**Contribution:** 2 fair
**Rating:** 5
**Confidence:** 4

**Summary:**

This submission introduces a new dataset of long-form, conceptually diverse videos, designed for tasks in video summarization and captioning. The authors offer statistics regarding the dataset, as well as performance evaluations of human annotators and baseline models using this data. Additionally, the paper discusses intriguing methodologies employed in dataset annotation and the quality assurance (QA) process.

**Strengths:**

1- The introduction of a diverse, high-quality dataset. This new dataset, with its meticulous annotation, has the potential to significantly benefit the research community.
2- The paper provides detailed methodologies for sampling from extensive video collections and data cleaning techniques. These practices could be invaluable for future data collection efforts by other researchers.
3- The paper presents intriguing findings comparing N-gram versus model-based metrics, as well as insights into the consistency of human performance in summarization and captioning tasks.

**Weaknesses:**

1- The paper lacks a comprehensive table or reference comparing the characteristics of this new dataset with existing datasets. Such a comparison is crucial for understanding the unique advantages and potential applications of the new data, guiding researchers in deciding why and when to use it.
2 - The paper provides insufficient qualitative examples of the dataset, leaving readers without a clear understanding of the data's content and quality. More illustrative examples would enhance comprehension and showcase the dataset's practical applications.
3- The paper's writing style is challenging to follow. Improving the clarity and flow of the writing would make the content more accessible and comprehensible to readers.
4- Combining the data on human and baseline method performances into one table would make comparisons easier and clearer.
5- The paper doesn't clearly explain the novelty or rationale behind the chosen baseline model (SimCSR). More details on why this particular design was selected, its effectiveness as a baseline, and its performance on similar datasets would be beneficial for understanding its relevance and comparison to other models.
6- The hallucination leveling seems to be subjective a subjective method.

**Questions:**

1- How reliable is youtube video chapter information? Why the authors use them and not annotating each video from scratch?
2- How do you measure the diversity of the data in this dataset?
3- Prompt engineering in building the training data can be a source of bias. How do you prevent that?

---

> ### Author Response · Authors · 2023-11-22
>
> We thank the reviewer for the comment. We plan on extensively revising our manuscript with a larger evaluation dataset (10K) and more extensive benchmarking and will be withdrawing our submission as a result.
> We take to heart the reviewer's feedback on the clarity and flow of the manuscript and will be revising the paper accordingly.
> We'd also like to address a few points raised by the reviewer for clarification.
> 1. Questions 1 and 3 raised by the reviewer both concern the training dataset. We'd like to clarify that the training dataset is not part of the released video-csr evaluation dataset and was included for completeness in describing our end-to-end SimCSR model. Nevertheless, to address the reviewer's question directly, it was not possible to secure enough human annotators to annotate each training video from scratch as there were 200K video clips in the train dataset. As a result, metadata information from videos was used to generate a large training dataset similar to techniques employed by contemporary works such as VAST-27M. As per prompt engineering being a source of bias in the training dataset, we sampled multiple captions/summaries for each prompt (i.e. each video clip) to hopefully reduce the bias in the model generated caption/summary. However, the paper was indeed lacking in a quantitative analysis of the bias in the model generated responses and we'll be working to improve upon this aspect of the work.
> 2. Evalution of hallucination: evaluation of hallucination is indeed a relatively subjective metric. However, in the absence of tried-and-true quantifiable measures that specifically gauge model hallucination, we believe that human evaluation is the most reliable alternative. However, to provide a more unbiased evaluation of model hallucination, we intend to increase the number of samples evaluated.

---

### Official Review · Reviewer_V9Rq · 2023-10-31

**Soundness:** 3 good
**Presentation:** 3 good
**Contribution:** 3 good
**Rating:** 6
**Confidence:** 4

**Summary:**

The paper introduces a new benchmark for both video captioning and video summarization. The dataset features a meticulously curated set of human annotations, designed to validate and test the capabilities of video-language models. This is particularly pertinent given the current trend of LLMs tackling vision tasks. The dataset will be invaluable for benchmarking the performance of Video LLMs and will serve as a useful resource for future research aiming to gauge advancements in the realm of video summarization and captioning. Additionally, the paper presents a method for summary retrieval, which is crucial for assessing the alignment between a model's visual and language features. The authors also conduct a comprehensive analysis of appropriate metrics for evaluating text generation tasks. The dataset's quality is ensured through rigorous curation and annotation validation, with multiple annotators employed and their annotation quality being assessed in various ways. To conclude, the paper offers a competitive baseline for the outlined tasks, rivaling some of the existing Video LLM works.

**Strengths:**

1. The paper introduces an important benchmark that is especially pertinent for the current generation of Video LLMs.
2. A thorough and insightful analysis has been conducted to ensure that the chosen metrics align closely with human judgment.
3. The authors have demonstrated great attention to detail in their exhaustive analysis of annotation quality.
4. The inclusion of retrieval as an integral component of the benchmark is a commendable and innovative addition, further enhancing the utility of the framework.

**Weaknesses:**

The paper has some addressable weaknesses that I'd like the authors to clarify:
1. The paper doesn't provide a clear comparison with other benchmarks of a similar nature. It remains unclear why this benchmark is preferable over others for evaluating Video LLMs.
2. While the proposed benchmark in the paper is promising, the depth of its benchmarking appears limited. The authors have exclusively experimented with a single VideoLLM. Yet, as they've acknowledged, several other models exist [D,E]. A more exhaustive benchmarking against current methodologies would significantly strengthen the paper's contributions. I'd also suggest adapting some image-based methods [A,B,C] using a straightforward sliding window inference. Such an addition would offer a broader and more credible spectrum of methods, encouraging the community to adopt the benchmark.
    - Why is the performance of Video-LLava on the Captioning task not reported in Table 5?
    - Why is there an absence of other methods in the retrieval task assessment?
3. The paper's method is very simple and heavily based on VideoCoCa. Additionally, it does not perform better than Video-LLAMA. It is clearly not the strongest contribution of the paper.


[A] Zhu, D., Chen, J., Shen, X., Li, X., & Elhoseiny, M. (2023). Minigpt-4: Enhancing vision-language understanding with advanced large language models. arXiv preprint arXiv:2304.10592.

[B] W Dai, J Li, D Li, AMH Tiong, J Zhao, W Wang, B Li, P Fung, S Hoi. InstructBLIP: Towards General-purpose Vision-Language Models with Instruction Tuning. NeruRIPS 2023.

[C] Wu, S., Fei, H., Qu, L., Ji, W., & Chua, T. S. (2023). Next-gpt: Any-to-any multimodal llm. arXiv preprint arXiv:2309.05519.

[D] Li, K., He, Y., Wang, Y., Li, Y., Wang, W., Luo, P., ... & Qiao, Y. (2023). Videochat: Chat-centric video understanding. arXiv preprint arXiv:2305.06355.

[E] Chen, J., Zhu, D., Haydarov, K., Li, X., & Elhoseiny, M. (2023). Video chatcaptioner: Towards the enriched spatiotemporal descriptions. arXiv preprint arXiv:2304.04227.

[F] Maaz, M., Rasheed, H., Khan, S., & Khan, F. S. (2023). Video-ChatGPT: Towards Detailed Video Understanding via Large Vision and Language Models. arXiv preprint arXiv:2306.05424.

**Questions:**

1. **Benchmark Comparison & Importance for VideoLLMs**: How does the introduced method compare against current video summarization/captioning benchmarks? Particularly for VideoLLMs, why is this benchmark needed? Point out limitations of current benchmarks and illustrate how this new one fills them or offers a better evaluation.

2. **Exclusivity of Video-llava in the Benchmark (Weakness 2)**: The paper predominantly highlights the evaluation with Video-llava. Could you explain the reason for focusing solely on this model and not incorporating other pertinent methods? Diversifying the benchmark with multiple methods would enrich the evaluation and make the results more universally applicable.

3. **Necessity & Novelty of the Proposed Method (Weakness 3)**: The necessity of the proposed method is currently under-explained. Can you provide a detailed argument about the unique advantages or features that your approach introduces, especially when compared with existing methods? Given the apparent simplicity and heavy reliance on VideoCoCa, as well as its underwhelming performance in comparison to Video-LLAMA, it would be beneficial to understand the distinct value proposition of your method.

**Citation Correction**: It's pivotal to credit research correctly. Notably, the ActivityNet dataset was introduced by Caba Heilbron et al. in 2015 [H], not by Krishna et al. in 2017 as cited in the paper. Such discrepancies can compromise the credibility of the paper, and I'd urge a meticulous review to avoid similar oversights.

[H] Heilbron, F. C., Escorcia, V., Ghanem, B., & Niebles, J. C. (2015). ActivityNet: A large-scale video benchmark for human activity understanding. In Proceedings of the IEEE conference on Computer Vision and Pattern Recognition (pp. 961-970).

---

> ### Author Response · Authors · 2023-11-22
>
> We thank the reviewer for the detailed feedback. We plan on extensively revising our manuscript with a larger evaluation dataset (10K) and more extensive benchmarking and will be withdrawing our submission as a result.
> We acknowledge the fact that, due to time constraints, benchmarking in the current paper is limited in terms of its extensiveness against current SOTA models. The proposed SimCSR model was solely included to serve as a baseline for the video-csr benchmark and we plan on potentially replacing the SimCSR model in favor of a more complete evaluation against existing video-to-text models.
> With respect to comparison of our evaluation dataset with other datasets, we'd like to reiterate that the current dataset is the first evaluation dataset where the textual description of video content is paragraph-length. As the reviewer pointed out, with the advent of LLMs that are capable of generating pages of texts, long-form image-to-text and video-to-text generation tasks are within reach of the latest video-language models. However, to the best of our knowledge, such evaluation benchmark is currently missing in the literature. Additionally, as we explored in the manuscript, the gap between syntactic and semantic similarities widens dramatically as the length of the textual descriptions increases. As such, we believe the video-csr dataset defines a new type of evaluation benchmark that fulfills a need of the current multimodal research effort.